# Endophytic Fungal Community of *Stellera chamaejasme* L. and Its Possible Role in Improving Host Plants’ Ecological Flexibility in Degraded Grasslands

**DOI:** 10.3390/jof9040465

**Published:** 2023-04-12

**Authors:** Wenting Tang, Weijun Gong, Ruitong Xiao, Wenqin Mao, Liangzhou Zhao, Jinzhao Song, Muhammad Awais, Xiuling Ji, Haiyan Li

**Affiliations:** 1Life Science and Technology & Medical Faculty, Kunming University of Science and Technology, Kunming 650500, China; 2College of Life Sciences, South China Agricultural University, Guangzhou 510642, China

**Keywords:** endophytic fungi, *Stellera chamaejasme* L., community composition, plant growth promotion, degraded grasslands

## Abstract

*Stellera chamaejasme* L. is a widely distributed poisonous plant in Chinese degraded grasslands. To investigate the role of endophytic fungi (EF) in *S. chamaejasme’s* quick spread in grasslands, the endophytic fungal community of *S. chamaejasme* was studied through culture-dependent and culture-independent methods, and the plant-growth-promoting (PGP) traits of some culturable isolates were tested. Further, the growth-promoting effects of 8 isolates which showed better PGP traits were evaluated by pot experiments. The results showed that a total of 546 culturable EF were isolated from 1114 plant tissue segments, and the colonization rate (CR) of EF in roots (33.27%) was significantly higher than that in shoots (22.39%). Consistent with this, the number of specific types of EF was greater in roots (8 genera) than in shoots (1 genus). The same phenomenon was found in culture-independent study. There were 95 specific genera found in roots, while only 18 specific genera were found in shoots. In addition, the dominant EF were different between the two study methods. *Cladosporium* (18.13%) and *Penicillium* (15.93%) were the dominant EF in culture-dependent study, while *Apiotrichum* (13.21%) and *Athelopsis* (5.62%) were the dominant EF in culture-independent study. PGP trait tests indicated that 91.30% of the tested isolates (69) showed phosphorus solubilization, IAA production, or siderophores production activity. The benefit of 8 isolates on host plants’ growth was further studied by pot experiments, and the results indicated that all of the isolates can improve host plants’ growth. Among them, STL3G74 (*Aspergillus niger*) showed the best growth-promotion effect; it can increase the plant’s shoot and root dry biomass by 68.44% and 74.50%, respectively, when compared with the controls. Our findings revealed that *S. chamaejasme* has a wide range of fungal endophytic assemblages, and most of them possess PGP activities, which may play a key role in its quick spread in degraded grasslands.

## 1. Introduction

Grasslands are one of the most important types of terrestrial ecosystems on earth. They have the ecological functions of preventing impacts from wind, fixing sand, maintaining water and soil, purifying air and maintaining biodiversity [1]. The grassland biome accounts for approximately one third of the land area of China, and provides great ecological and economic value to the country [2]. However, 90% of the natural grassland has deteriorated to various degrees during the past 50 years due to intense human disturbance, especially overgrazing [3,4]. The increasing abundance of poisonous plants is one important indicator of grasslands degradation. These plants often have strong adaptive capacity and produce secondary compounds that are toxic to livestock, wild herbivores, and humans [5,6,7]. Therefore, it is imperative for policy makers and herders to manage and control the increasing number of poisonous plants on degraded grasslands.

*Stellera chamaejasme* L. is a poisonous plant that inhabits a wide range of altitudes from 130 to 4200 m, including a broad area from southern Russia to southwest China and the western Himalayas [8]. They are the most dominant noxious weeds in degraded grasslands [5,9]. *S. chamaejasme* patches tend to create islands of fertility (i.e., greater soil nutrient availability). Sun et al. [10] found that the total nutrients, microbial biomass, cycling of carbon and nitrogen, and availability of nitrogen in *S. chamaejasme* patches were greater than elsewhere. Researchers have explained that *S. chamaejasme* has strong ecological flexibility and is a highly competitive species in degraded grasslands, reflecting its relationship with the soil, competition with other plants, and harm to livestock [7,11,12,13].

Endophytes are microorganisms that live inside internal tissues of the host plant without causing symptoms of disease [14]. Endophytes make host plants more tolerant to various biotic and abiotic stresses, such as pathogens, drought, salinity, nutrient deficiency, and metal toxicity [15]. A variety of studies have shown that endophyte-infected plants can be more tolerant of low nutrient inputs. The presence of fungal endophytes improves the survival of *Lolium perenne* in low fertility soils by increasing root growth, metabolic activity and absorption of nutrients [16]. Under low phosphorus (P), fungal endophytes altered how plant biomass responded to soil nitrogen (N) and how plant traits responded to soil phosphorus (P) and showed improved forage quality [17]. Endophytic fungi (EF) can enhance plant growth and crop yields through P solubilization, siderophore and indole-3-acetic acid (IAA) production, etc. Baron et al. [18] demonstrated that inoculation with the fungi *Purpureocillium lilacinum*, *Purpureocillium lavendulum* and *Metarhizium marquandii*, which can produce IAA and solubilize P, could promote the growth of maize, bean and soybean. The degraded grassland is not suitable for the growth of most plants due to its low content of soil organic matter and water. However, *S. chamaejasme* can proliferate and become the dominant plant, which may be related to endophytic fungal benefits. Jin et al. found that the endophytic fungal communities of *S. chamaejasme* differed significantly within plant tissues and growth stages in both cultivation-dependent and cultivation-independent contexts [19,20]. However, the role of endophytic fungi in *S. chamaejasme* in degraded grasslands is unknown. Therefore, in the present study, the endophytic fungal community of *S. chamaejasme* was investigated through culture-dependent and independent methods, and the growth-promoting ability of some culturable isolates were evaluated. Furthermore, the benefit of 8 isolates, which showed better plant-growth-promoting (PGP) traits, was evaluated by pot experiments.

## 2. Materials and Methods

### 2.1. Plant Material and Treatments

The sampling sites were located in Xiaozhongdian town, Shangri-La county, Diqing Tibetan Autonomous Prefecture, Yunnan province, Southwest China (27°49′ N, 99°49′ E). The average elevation is 3350 m. With a protracted frost season, short spring and fall, a long winter, and no distinguishable summer, the climate is cold and temperate. The average annual temperature is 6.3 °C, and there are 800–1000 mm of rain every year [21]. *S. chamaejasme* is the dominant species in the degraded grasslands of Xiaozhongdian town. The samples were collected in July 2020 and placed into sterile plastic bags, brought back to the lab and further processed within 24 h.

The plants were washed with running tap water to remove the adhering soil particles and other contaminants. Surface sterilization was carried out by sequentially dipping the fragments into 75% ethanol for 2 min, rinsing with sterile distilled water, followed by treatment with 5% sodium hypochlorite for 2 min and finally 3–5 rinses with sterile distilled water [22]. The surface sterilized fragments were dried on sterilized filter paper, and the effect of the surface sterilization process was confirmed by making imprints of disinfected plant fragments in Petri dishes containing PDA (potato dextrose agar); the absence of any fungal growth indicated an effective surface sterilization [23]. Some parts of the sterilized plants were used for fungal endophyte isolation, and the remainder were stored at −80 ℃ for further DNA-based analyses. For each analysis, three biological replicates were made.

### 2.2. Culturable Endophytic Fungal Community Analysis

The surface sterilized plants were cut into segments (about 5 × 5 mm), then, 200 root segments and 200 shoot segments (100 stems and 100 leaves) from each sample were selected randomly and placed on Petri dishes containing PDA supplemented with 0.5 g/L streptomycin sulphate. The plates were incubated at 25 °C and checked every other day for 45 days; fungi growing out of the plant tissues were transferred to fresh PDA plates [24]. All of the isolates were deposited at the Medical School of Kunming University of Science and Technology.

Fungal endophytes identification was based on their morphological characteristics as well as the internal transcribed spacer (ITS) region analysis [25,26]. For ITS region analysis, the CTAB extracted genomic DNA from fungal isolates were amplified using the primers ITS1 and ITS4 [27]. Following the manufacturer’s instructions, the PCR products were purified using the Cyclepure Kit (Bioteke: Beijing, China), and then delivered to Sangon Biotech Co., Ltd. (Shanghai, China) for sequencing. Finally, the sequences obtained in this study were uploaded to GenBank database (http://www.ncbi.nlm.nih.gov/, accessed on 15 January 2023) and the similarities of them with the published sequences in GenBank database were determined by BLAST.

The colonization frequency (CF) of EF was calculated as the total number of segments colonized by endophyte divided by the total number of incubated segments. The relative species frequency (RF) of EF was calculated as the number of isolates of one species divided by the total number of isolates [28].

The Shannon diversity index (H′) of endophytic fungi was calculated according to the formula: $H^{\prime }=-\sum_{i=1}^{k}P_{i}\times \ln P_{i}$, where k is the total number of fungal species, and P_i_ is the proportion of individuals that species i contributes to the total [29]. Evenness values were calculated following Pielou’s Evenness Index. Pielou’s Evenness Index J = H′/log (S), where S is the number of species (species richness) [30].

### 2.3. Culture-Independent Endophytic Fungal Community Analysis

Each surface-sterilized plant sample was homogenized in a sterile mortar with liquid nitrogen. Then, 0.2 g of homogenized powder was used to extract the total genomic DNA by MoBio PowerSoil^®^ DNA Isolation Kit (MO BIO Laboratories, Inc., Carlsbad, CA, USA) following the manufacturer’s protocol. The extracted DNA was verified by electrophoresis on a 1.5% (*w*/*v*) agarose gel, and the qualified DNA was stored at −20 ℃ for subsequent analyses. Amplification of the fungal ITS1 gene region was performed using primer ITS1F (5′-CTTGGTCATTTAGAGGAAGTAA-3′) and ITS2R (5′-GCTGCGTTCTTCA TCGATGC-3′); the amplification product was about 435 bp. PCR reaction conditions were as follows: pre-denaturation at 98 °C for 3 min; denaturation at 94 °C for 45 s; annealing at 55 °C for 45 s; extension at 72 °C for 45 s; 30 cycles; extension at 72 °C for 7 min. The PCR products were recovered by 1.5% agarose gel and purified by AxyPrep DNA Gel Extraction Kit, eluted with 1.5% agarose electrophoresis. Quanti-Fluor™-St (Promega, Madison, WI, USA) blue fluorescence quantitative system was used for detection of PCR products. Purified amplicons were pooled in equimolar and paired-end sequences (2 × 300), according to Illumina, San Diego, USA standard operating procedures [31]. All steps were implemented at Shanghai Majorbio Bio-pharm Technology Company (Shanghai, China). The Illumina sequencing data are available in the NCBI Sequence Read Archive (SRA) repository under BioProject accession number PRJNA904100.

The original sequence was quality controlled by Trimmmatic software and spliced by Flash software. After trimming barcode sequences and adapter region, quality filtering settings were as follows: (1) minimum twenty quality score of the sequence read, no truncated reads shorter than 50 bp; (2) mismatch ratio between overlaps less than 0.2, the length of overlaps more than 10 bp, no reads that could not be assembled; (3) two primer base mismatches allowed, no reads containing ambiguous characters [32]. After further denoising the sequence within each sample using a pre-clustering algorithm, the optimized sequences were clustered into operational taxonomic units (OTUs) based on 97% pairwise identity using UPARSE (v7.0, http://www.drive5.com/uparse/, accessed on 15 January 2023). A representative sequence for each OTU was taxonomically classified and annotated according to the UNITE database (Release8, https://unite.ut.ee/, accessed on 15 January 2023) using the RDP Classifier algorithm (https://sourceforge.net/projects/rdp-classifier/, accessed on 15 January 2023). The species richness and diversity indexes including the Shannon, Simpson, Chao1 and ACE index of all the samples were analyzed by the Mothur software [33].

### 2.4. Growth-Promoting Traits Test

The indole-3-acetic acid (IAA), siderophore producing, and phosphate solubilizing activities of some culturable endophytes were tested. The endophytes were inoculated into specific liquid media and incubated for 10 days at 28 °C, and the content of IAA, siderophore and phosphate was determined according to the following methods: IAA was quantified using the Salkowski reagent [34], siderophore was detected by the CAS assay under Fe^3+^ limiting conditions [35], and phosphate solubilization capacity was tested by the method of Jain et al. [27]. All the experiments were performed in triplicates.

### 2.5. Pot Experiments

*L*. *perenne* is one of the widely cultivated forage plants and often selected as a model species for stress studies [36]. Therefore, *L. perenne* were used in pot experiments in the present study to evaluate the benefit of fungal endophytes which showed better growth- promoting traits. The seeds of *L. perenne* were sterilized as above. The fungal endophytes (Table 1) were cultured on PDB medium at 28 °C for 4–7 days, and 500 mg of the mycelia were collected, cut into pieces (1–3 mm) with sterile scissors, and re-suspended in 300 mL sterile distilled water. The sterilized seeds were divided into 2 portions randomly. Portion I was immersed in fungal suspension for 1 h for fungal inoculation (E+) [37]. Portion II was immersed in sterile water as the control (E-). Then the E+ and E- seeds were sown in pots, which contained 600 g autoclaved mixture of field soil with perlite (7:3, vol./vol.) and 5 g Ca_3_(PO_4_)_2_, respectively. Three pots per treatment were used, and 10 seeds were sown in each pot. After the first pair of true leaves appeared, 5 seedlings were kept in each pot [34]. The pots were arranged randomly and kept under artificial plant light (16:8 h light/dark cycle). The plants were watered with autoclaved water every 2–3 days according to the requirement. The plants were inoculated (sprayed with the mycelial suspension/sterile water) again as above on the 10th day. After 36 days culture, the plants were harvested and the height, root length and dry biomass were tested.

### 2.6. Statistical Analysis

All statistical analyses were performed with SPSS software ver. 11.5. The differences among the 8 endophytes inoculated were compared by the Duncan’s multiple-range test. Venn diagrams were drawn using TBtools. All the data were presented as mean ± standard deviation of three replicates.

## 3. Results

### 3.1. Culturable Endophytic Fungal Community

A total of 546 fungi were isolated from 1114 tissue segments, of which 348 were from roots and 198 from shoots (Table 2). The colonization rate (CR) of EF of the whole plant was 30.16%, and the CR of the root was 33.27%, which was significantly higher than that of the shoot (22.39%) (*p* < 0.05, *t*-test) (Table 2). The isolates were divided into 27 genera based on morphological characteristics and ITS sequence analysis. Among them, 18 genera (30.23%) co-existed in both shoots and roots, while 1 genus was found uniquely associated with shoot tissues, and 8 genera were only isolated from root tissues (Figure 1A). The dominant EF of all samples were *Cladosporium*, *Penicillium*, *Phaeosphaeriaceae* and *Plenodomus*, showing a relative frequency (RF) of 18.13%, 15.93%, 8.61% and 7.88%, respectively. However, the dominant EF of shoots showed a little difference from that of roots. In shoots, the dominant EF were *Cladosporium* (27.05%), *Penicillium* (13.42%) and *Phaeosphaeriaceae* (12.69%), while in roots the dominant EF were *Penicillium* (17.65%), *Cladosporium* (14.15%) and *Ilyonectria* (10.49%) (Figure 2A). Table 3 shows that all diversity indices of the roots (H′: 2.55 ± 0.21; Evenness: 0.86 ± 0.05; chao1: 24.89 ± 5.17; Simpson’s indices: 0.90 ± 0.03) were higher than those of the shoots (H′: 2.13 ± 0.28; Evenness: 0.83 ± 0.04; chao1: 15.54 ± 5.51; Simpson’s indices:0.84 ± 0.06); however, there was no significant difference (*t*-test, *p* > 0.05). The rDNA ITS sequences of the fungi subjected to molecular identification in this study were deposited in GenBank (Accession numbers are OQ438905, OQ438904, OQ438903, OQ438901, OQ438885, OQ438800, OQ438651, OQ438617, OQ438432 and OQ438427).

### 3.2. Culture-Independent Endophytic Fungal Community

A dataset was developed that consisted of 614,278 filtered high-quality and classified unique fungal ITS1F_ITS2R gene tags with the average length of 258 bp (Table 4). All of these fungal sequences were grouped into 248 OTUs. Among them, 35 OTUs appeared in both roots and shoots, while 166 OTUs were only found in roots, and 47 OTUs were only found in shoots (Figure 1B). The OTU Rank-Abundance curve indicated that the endophytes of roots were relatively greater and more abundant than those of shoots (Figure 3A).

The OTUs were assigned into 7 phyla, 20 classes, 47 orders, 99 families, 144 genera and 175 species. Ascomycota was found to be the most abundant (47.4%) across all samples analyzed, followed by Basidiomycota (23.85%). More OTUs remained unclassified in shoots (49.24%) in comparison with roots (0.42%). At the genus level, it was found that roots were dominated by *unclassified_p__Ascomycota* (32.30%) OTUs, then followed by *unclassified_f__Herpotrichiellaceae* (20.08%), *Athelopsis* (13.22%) and *Lachnum* (7.19%). *Apiotrichum* (21.01%) was the dominant endophyte in shoots, then followed by *Plenodomus* (3.88%), *Cutaneotrichosporon* (3.44%) and *Aspergillus* (2.4%) (Figure 2B).

The computational analysis of α-diversity estimated the richness and diversity of two plant tissues at OTU cut-offs of 0.03 distance units (Table 4). Among them, Chao1 and Ace estimated the minimum number of OTUs, and inverse Simpson and Shannon diversity index indicated the richness of the communities. It was found that the Chao1, Ace and Simpson diversity of shoots EF were lower than those of roots (with significant differences, *p* < 0.05, for Chao1, Ace) (Table 4). However, the Shannon indices of shoots EF were higher than those of roots (no significant difference, *p* > 0.05). Beta diversity analysis indicated that endophytic fungal species diversity between the shoot and root was significantly different (Figure 3B), as was observed in the circos graph (Figure 3C) and heat map (Figure 4).

### 3.3. PGP Traits of Fungal Endophytes

Sixty-nine fungal isolates from 22 genera were randomly chosen for the PGP traits test. The result indicated that 91.30% of the tested isolates exhibited phosphorus solubilization, IAA production, or siderophores production activity, whereas only 15.94% of the tested isolates displayed all three PGP activities.

Phosphate solubilization tests indicated that 48 isolates (69.57%) from 20 genera can solubilize P with the range of 22.67 to 296.33 mg/L. Among them, the isolate STL3G74 (*Aspergillus*; 296.33 mg/L) showed the highest P solubilization, followed by STL3G8 (*Plenodomus*; 292.17 mg/L), LT1G13 (*Porostereum*; 230.18 mg/L) and STL1G60 (*Ilyonectria*; 195.67 mg/L) (Figure 5C).

IAA synthesis tests indicated that 37 isolates (53.62%) from 14 genera can produce IAA. IAA production ranged from 22.94 mg/L by the isolate LT3S41-1 (*Alternaria*) to 208.42 mg/L by STL3G74 (*Aspergillus*). The isolates STL1G60 (*Ilyonectria*), LT2S21 (*Gibberella*) and LT2G53 (*Ilyonectria*) also showed better IAA producing capability with 136.97, 124.28 and 106.62 mg/L, respectively (Figure 5A).

Of the 69 isolates tested, 26 (27.07%) could produce siderophores, since their percentages of siderophore units (SU) were more than 10%; the %SU quantified by these isolates ranged from 15.23% to 87.64%. Among them, 10 isolates produced SU ≥ 50%; the strain STL3G74 (*Aspergillus*) was the best with 83.38% SU production, followed by LT2G30 (*Trametes*) and LT2S36 (*Phaeosphaeriaceae*) with 80.73% and 80.36% SU (Figure 5B).

### 3.4. Pot Experiments

Eight isolates (Table 1) which showed better PGP traits were chosen to conduct pot experiments. Results showed that all 8 isolates can improve *L. perenne* growth. They significantly increased the shoot dry biomass (*p* < 0.05), except the isolate LT3G35 (*p* > 0.05). However, only the isolates STL3G74, LT2S21, STL1G60 and LT2G30 significantly improved the root dry biomass (*p* < 0.05). Some of them enhanced the shoot and root length, but there was no significant difference between E+ and E- (*p* > 0.05) (Figure 6A). Among all isolates, STL3G74 showed the best growth-promotion effect. When compared with E-, the shoot length, shoot and root dry biomass of E+ were increased 4.69%, 68.44% and 74.50%, respectively (Figure 6B).

## 4. Discussion

To discover the role of endophytic fungi in *S. chamaejasme’s* strong competitiveness in degraded grasslands, the endophytic fungal community of *S. chamaejasme* and its growth-promoting function were studied. The results showed that the number of fungi from roots was higher than that from shoots, whether in culture-dependent or independent conditions (Table 2 and Table 4). In agreement with this, Jin et al. also found that the number of EF from *S. chamaejasme* roots was clearly higher than that from shoots both in culture-dependent [19] and independent [20] studies. Similarly, previous investigations in *Aristolochia chilensis* of an arid ecosystem found that fungal isolates from root tissues were almost two times higher than those from shoot tissues [38]. The calculation of diversity indices provided further evidence for community compositional differences between tissue types, suggesting low species turnover between shoot and root tissues (Table 3 and Table 4). These results agree with previous findings that community composition was strongly determined by plant tissue type [39,40,41]. Similarly, the total colonization frequency and species richness of EF in the root were higher than those in the shoot. It has been suggested that the longer life and higher biomass of roots in comparison to shoots might be a relevant factor influencing this pattern [12,38,42]. Further studies analyzed the high diversity of root endophytic fungi, which may be due to the fact that there are more litter residues (fallen leaves, branches, and dead roots) and rhizosphere exudates at the contact interface between roots and soils. They can give the rhizosphere and root fungi enough readily available carbon, nitrogen, and trace elements, which in turn helps to increase their richness [43,44].

Seven fungal phyla were recovered in culture-independent study, while only 3 fungal phyla were detected in culture-dependent study. Ascomycota was found to be the most common EF in plants by both culture-dependent (95.97%) and culture-independent (47.43%) methods. This is consistent with previous findings that Ascomycota are the most prevalent group within plant tissues [45,46,47]. In addition, the endophytic assemblages associated with *S. chamaejasme* exhibited a certain degree of tissue specificity: 95 genera were only found in the root, and 18 genera were only found in the shoot by culture-independent methods. Similarly, 8 genera were only found in the root, and 1 genus was only found in the shoot in culture-dependent study (Figure 1) [20]. Moreover, culture-dependent study showed that *Cladosporium* (18.13%) was the most dominant genus; however, the dominant genus in culture-independent study was *Apiotrichum* (13.21%), and *Cladosporium* only showed 1.49% (Figure 2). These results suggest that to understand the endophytic community comprehensively, a combination of both culture-dependent and culture-independent methods is necessary.

Previous studies have demonstrated that microbial endophytes directly promote plant growth through phosphorus mobilization, plant hormones production, etc. [48]. Furthermore, these microbial endophytes may provide essential iron to the plant for performing various cell functions [48,49]. In the present study, 91.30% of the tested isolates exhibited phosphorus solubilizing, IAA, or siderophores producing activity, and 15.94% of them possessed all three PGP activities (Figure 5). Jain et al. [27] found that all tested endophytes of *Arnebia euchroma* variably possessed one or more important PGP traits. Consistent with this, the isolate STL3G74, which showed the best PGP activities, showed the best host plant growth-promoting ability in pot experiments. Compared with un-inoculated plants, the shoot length, shoot and root dry biomass of plants inoculated with STL3G74 increased by 4.69%, 68.44% and 74.50%, respectively (Figure 6). STL3G74 was identified to be *Aspergillus niger.* Previous studies indicated that *A. niger* is a multifunctional fungus capable of phosphate solubilization, potassium solubilization, organic acid, siderophore and phytohormone production [50,51,52]. Plants fertilized with phosphate solubilized by *A. niger* showed enhanced growth and P uptake. For example, Araújo et al. [53] found that an isolate of *A. niger* can promote the growth of coffee seedlings in a substrate with no P limitation. Similarly, a previous study indicated that inoculation of *A. niger* can increase the production of shoot fresh mass of lettuce (61%), kale (40%), scarlet eggplant (101%), watermelon (38%), melon (16%), pepper (92%), and tomato (42%) [51]. Baron et al. [18] found that maize (*Zea mays*) inoculated with *A. sydowii* accumulated significantly higher amounts of P in a field study even when receiving lower fertilization doses. Moreover, our previous study showed that 93.65% of the tested bacterial endophytes (63) exhibited nitrogen-fixing, IAA-synthesizing, phosphorus or potassium solubilizing capacity and that ST3CS3 could significantly improve host plants’ growth and increase their nitrogen and chlorophyll content [54]. Therefore, we suggest that the strong competitiveness of *S. chamaejasme* may in part be due to its possession of high ratios of plant-growth-promoting endophytic fungi and bacteria. However, research on different endophyte taxa and the related scientific disciplines have largely developed separately, and studies on bacterial and fungal interactions and their importance are lacking. Studying the combined effects of endophytic fungi and bacteria may better explain *S. chamaejasme*’s competitiveness in degraded grasslands in the future.

## Figures and Tables

**Figure 1 jof-09-00465-f001:**
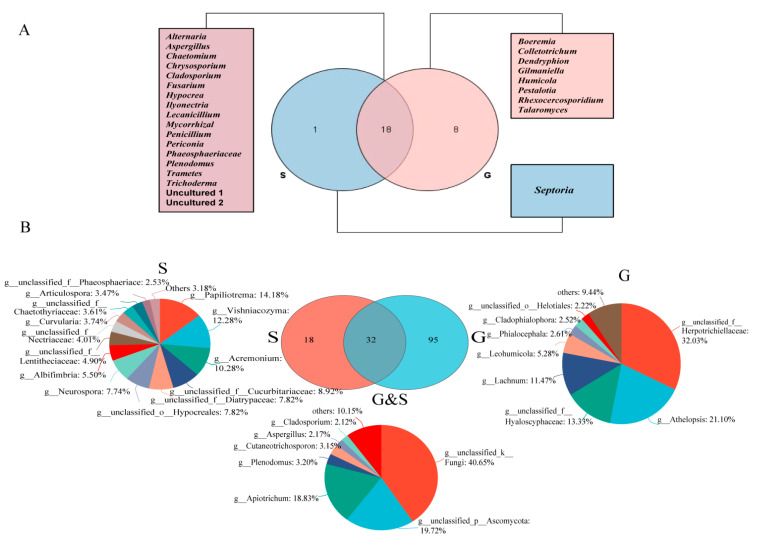
The unique and shared endophytic fungi from different tissues of *S. chamaejasme* at genus level. (**A**): culture-dependent endophytic fungi; (**B**): culture-independent endophytic fungi. The fungi with a percentage below 2% were grouped as ‘others’. S = Shoot; G = Root; G & S = Root & Shoot.

**Figure 2 jof-09-00465-f002:**
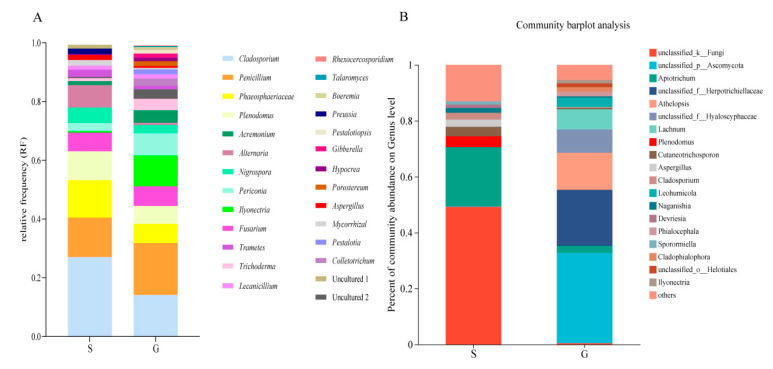
The relative abundance of endophytic fungi of *S. chamaejasme* at genus level. (**A**): culture-dependent endophytic fungi; (**B**): culture-independent endophytic fungi. The fungi with a relative abundance below 0.1% were grouped as ‘others’. S = Shoot; G = Root.

**Figure 3 jof-09-00465-f003:**
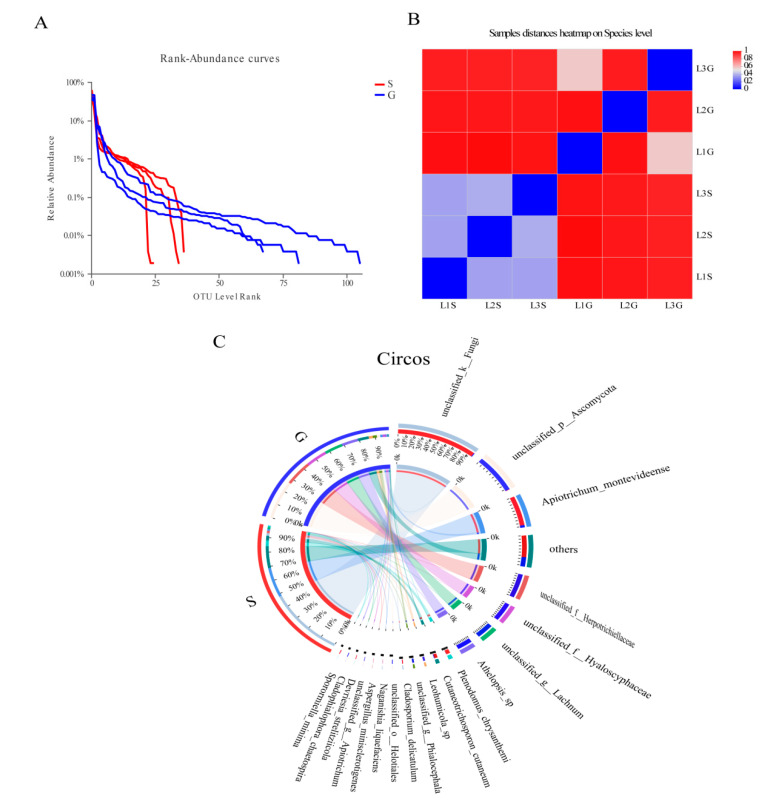
The diversity of culture-independent endophytic fungi between the shoot and root of *S*. *chamaejasme.* (**A**): The rank curve based on OTU abundance; (**B**): Beta diversity heatmap based on Unweighted UniFrace distance. Different color indicates difference in samples of species diversity; (**C**): Circos graph. The left semicircle represents the species composition of each group. The right semicircle indicates the distribution of each species in the different groups. S = Shoot; G = Root.

**Figure 4 jof-09-00465-f004:**
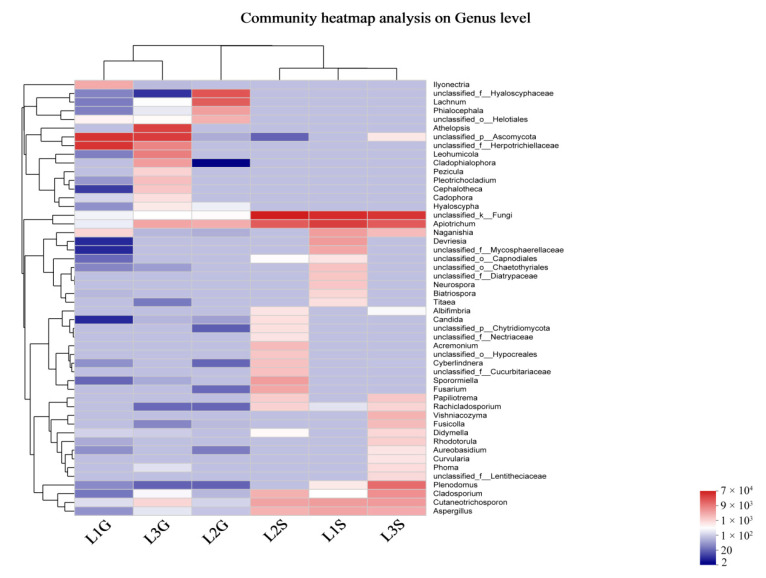
Heat maps of the relative abundance of culture-independent endophytic fungi in *S. chamaejasme* at genus level, based on an analysis of the first 50 most abundant genera. Different color indicates difference in relative abundance (Log10) of the taxa in the samples (red means high relative abundance).

**Figure 5 jof-09-00465-f005:**
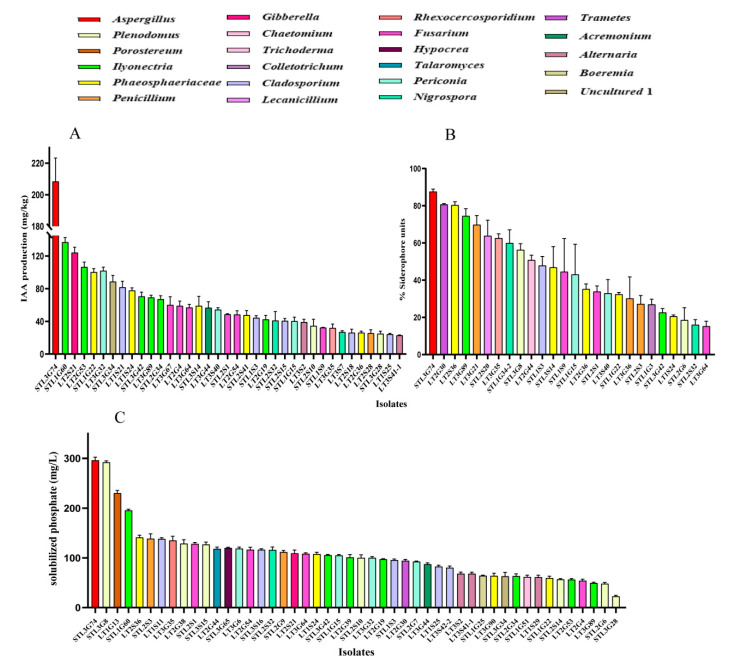
The IAA production (**A**), siderophores production (**B**) and phosphorus solubilization (**C**) activity of fungal endophytes of *S. chamaejasme*. Error bars represent standard deviation (*n* = 3). The same colors indicate they belong to the same genus.

**Figure 6 jof-09-00465-f006:**
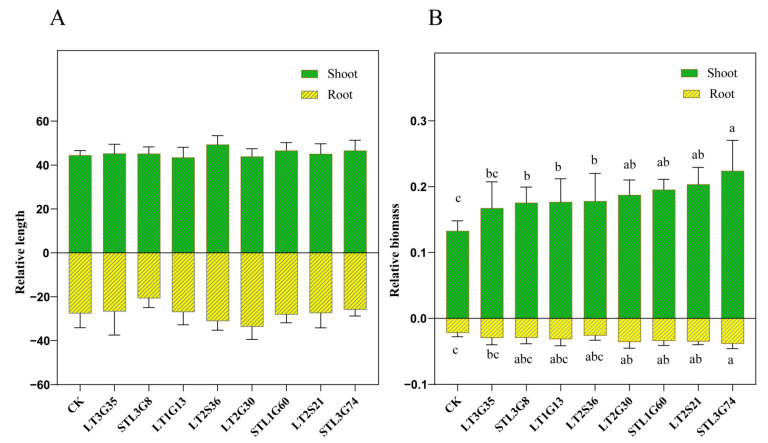
The relative length (**A**) and relative biomass (**B**) of *Lolium perenne* inoculated with different isolates. Different letters in each column denote that mean values are significantly different (*p* < 0.05), means ± SD (*n* = 10).

**Table 1 jof-09-00465-t001:** The isolates used in pot experiments and their plant growth promoting raits.

Isolate	Morphotype	Solubilized Phosphate (mg/L)	IAA (mg/L)	SU%
STL3G74	*Aspergillus niger*	296.33	208.42	87.64
LT2S21	*Gibberella* sp.	109.30	124.28	-
STL1G60	*Ilyonectria robusta*	195.67	136.96	-
LT2G30	*Trametes* sp.	94.27	-	80.74
LT2S36	*Phaeosphaeriaceae* sp.	141.1	-	80.36
LT1G13	*Porostereum spadiceum*	230.18	-	-
STL3G8	*Plenodomus tracheiphilus*	292.78	-	56.31
LT3G35	*Rhexocercosporidium* sp.	135.03	32.05	62.56

Note: “-”, negative result.

**Table 2 jof-09-00465-t002:** The number and colonization rate (CR) of culture-dependent endophytic fungi (EF) of *S. chamaejasme*.

Sample	No. of Segments Plated (No. of Segments Colonized by EF)			No. of EF Isolated			CR (%)		
	G	S	Total	G	S	Total	G	S	Total
Ⅰ	188 (70)	195 (42)	383 (112)	125	67	192	37.23	21.54	29.24
Ⅱ	180 (53)	195 (50)	375 (103)	110	81	191	29.44	25.64	27.47
Ⅲ	181 (60)	175 (28)	373 (80)	113	50	163	33.15	16	21.45
Average	-						33.27 ± 3.9 a	22.39 ± 2.92 b	30.16 ± 3.25
Total	549 (183)	565 (120)	1114 (303)	348	198	546	-		

Note: Ⅰ, Ⅱ and Ⅲ represent 3 replicates. S = Shoot; G = Root. Different letters indicate a significant difference at *p* < 0.05, *t*-test.

**Table 3 jof-09-00465-t003:** Alpha diversity indices of culture-dependent endophytic fungi (EF) in tissues of *S. chamaejasme*.

Tissue	Sample ID	Diversity Indices of EF				
		Taxa (S)	H′	Evenness	Chao1	D
Shoot	S1	14	2.29	0.87	15.13	0.87
	S2	15	2.28	0.84	21.25	0.87
	S3	10	1.81	0.78	10.25	0.78
	Average	13 ± 2.65 a	2.13 ± 0.28 a	0.83 ± 0.04 a	15.54 ± 5.51 a	0.84 ± 0.06 a
Root	G1	21	2.78	0.91	25.00	0.93
	G2	17	2.37	0.84	19.67	0.88
	G3	21	2.50	0.82	30.00	0.89
	Average	19.67 ± 2.31 b	2.55 ± 0.21 a	0.86 ± 0.05 a	24.89 ± 5.17 a	0.90 ± 0.03 a

Note: Different letters in each column denote that mean values are significantly different (*p* < 0.05), means ± SD (*n* = 3).

**Table 4 jof-09-00465-t004:** The number of OTUs and alpha-diversity indices of culture-independent endophytic fungi of *S. chamaejasme* (distance < 0.03).

Tissue	Sample ID	Number of Sequences	OTU	α-Diversity			
				Simpson	Shannon	ace	Chao
Shoot	S1	119,019	82	0.30	1.733	25.75	82
	S2	124,023		0.38	1.715	35	68
	S3	109,998		0.21	2.207	37	108.45
	Average	117,680		0.30	1.885	32.59	86.15
Root	G1	98,814	201	0.43	1.143	82	82
	G2	53,288		0.30	1.663	68	68
	G3	109,136		0.24	2.036	108.45	108.5
	Average	87,079		0.32	1.61	86.15	86.17

## Data Availability

Not applicable.

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
