# Peer review of "Endophytic Fungal Community of Stellera chamaejasme L. and Its Possible Role in Improving Host Plants’ Ecological Flexibility in Degraded Grasslands"

_jof, 2023, doi:10.3390/jof9040465_

Round 1

Reviewer 1 Report

The manuscript describes and analyse the endophytic fungal community of Stellera chamaejasme, which may provide valuable information about its adaptation potential to degraded grasslands. Endophytic fungi were analysed both based on isolation and identification, and culture-independent method, and culturable fungi were further studied.  The results were correctly described, and visualized by the tables and figures.  Discussion part nicely connect the results with the previous knowledge on this field.

Minor comments:

Page 2, lanes 52: Sentences are two complicated. They should be fragmented, and re-formulated.

Table 1: Missing the description of the numbers in parenthesis.

Figure 1B: letters are too small, figures should be increased.

Page 11, Lane 325: delete one space between “in” and “culture-dependent”.

Author Response

Comment 1: Page 2, lanes 52: Sentences are two complicated. They should be fragmented, and re-formulated. 

Response: We have corrected it.

Comment 2: Table 1: Missing the description of the numbers in parenthesis.

Response It was our mistake, we have corrected it.

Comment 3: Figure 1B: letters are too small, figures should be increased.

Response: We have increased the size of letters in the picture.

Comment 4: Page 11, Lane 325: delete one space between “in” and “culture-dependent”.

Response: We have corrected it.

Reviewer 2 Report

The topic is interesting and important. Large number of experiments have been performed and are well structured and analyzed.  Minor corrections needed: 1) Line 65. Replace “lolium perenne” with “Lolium perenne”. 2) Line 178. Indicate how, with what tools and into how many large fragments you divided the mycelium. 3) Line 205. You write that “The plants were inoculated again as above on the 10th day”. I do not understand. In the beginning, you had seeds that were immersed in fungal suspension. Then you had seedlings. Did you immerse the seedlings too? 4) Line 205. Write 8.61%. 5) Lines 236, 259 and 263. Probably Table 3, not Table 4? 3) Line 359. In the discussion, you write that you have identified the species Aspergillus niger. It should also be written in the Methods and Results. Perhaps you also identified the other isolates mentioned in Table 4?

Author Response

Reviewer #2 comments to authors:

Comment 1:  Line 65. Replace “lolium perenne” with “Lolium perenne”.

Response: We have corrected it. 

Comment 2:  Line 178. Indicate how, with what tools and into how many large fragments you divided the mycelium.

Response: The fungal endophytes (Table 4) were cultured on PDB medium at 28 ℃ for 4–7 days, and the 500 mg mycelia were collected, cut into pieces (1-3 mm) with sterile scissors, and re-suspended in 300 mL sterile distilled water, respectively. The object is to make the fungal mycelium well-distributed. We have added it to Line 181 on Page 4 of the revised MS.

Comment 3: Line 205. You write that “The plants were inoculated again as above on the 10th day”. I do not understand. In the beginning, you had seeds that were immersed in fungal suspension. Then you had seedlings. Did you immerse the seedlings too?

Response: Endophytic fungi may disappear after a period of time and need repeated inoculation to ensure colonization in the plant. The plants were inoculated (sprayed with the mycelial suspension/sterile water) again as above on the 10th day. We have added it to Line 192, Page 4 of the revised MS.

Comment 4: Line 205. Write 8.61%. 

Response: We have corrected it.Comment 5: Lines 236, 259 and 263. Probably Table 3, not Table 4?

 Response: We have corrected it.

Comment 6: Line 359. In the discussion, you write that you have identified the species Aspergillus niger. It should also be written in the Methods and Results. Perhaps you also identified the other isolates mentioned in Table 4?

Response: I revised Table 4 and changed the methods and results accordingly
